# EINSTEIN VI: GENERAL AND INTEGRATED STEIN VARIATIONAL INFERENCE IN NUMPYRO

### ABSTRACT

Stein Variational Inference is a technique for approximate Bayesian inference that is gaining popularity since it combines the scalability of traditional Variational Inference (VI) with the flexibility of non-parametric particle based inference methods. While there has been considerable progress in development of algorithms, integration in existing probabilistic programming languages (PPLs) with an easy-to-use interface is currently lacking. EinStein VI is a lightweight composable library that integrates the latest Stein Variational Inference methods with the NumPyro PPL. Inference with EinStein VI relies on ELBO-within-Stein to support use of custom inference programs (guides), non-linear scaling of repulsion force, second-order gradient updates using matrix-valued kernels and parameter transforms. We demonstrate the achieved synergy of the different Stein techniques and the versatility of the EinStein VI library by applying it on examples. Compared to traditional Stochastic VI, EinStein VI is better at capturing uncertainty and representing richer posteriors. We use several applications to show how one can use Neural Transforms (NeuTra) and second-order optimization to provide better inference using EinStein VI. We show how EinStein VI can be used to infer novel Stein Mixture versions of realistic models. We infer the parameters of a Stein Mixture Latent Dirichlet Allocation model (SM-LDA) with a neural guide. The results indicate that Einstein VI can be combined with NumPyro's support for automatic marginalization to do inference over models with discrete latent variables. Finally, we introduce an example with a novel Stein Mixture extension to Deep Markov Models, called the Stein Mixture Deep Markov Model (SM-DMM), which shows that EinStein VI can be scaled to reasonably large models with over 500.000 parameters.

## 1 INTRODUCTION

Interest in Bayesian deep learning has surged due to the need for quantifying the uncertainty of predictions provided by machine learning algorithms. The idea behind Bayesian learning is to describe observed data $\mathbf{x}$ using a model with latent variable $\mathbf{z}$ (representing model parameters and nuisance variables, see e.g., Fig. 4a). The goal is then to infer a posterior distribution $p(\mathbf{z}|\mathbf{x})$ over latent variables given a model describing the joint distribution $p(\mathbf{z}, \mathbf{x}) = p(\mathbf{x}|\mathbf{z})p(\mathbf{z})$ following the rules of Bayesian inference:

$$p(\mathbf{z}|\mathbf{x}) = Z^{-1}p(\mathbf{x}|\mathbf{z})p(\mathbf{z})$$

where the normalization constant $Z = \int_{\mathbf{z}} p(\mathbf{x}|\mathbf{z})p(\mathbf{z})d\mathbf{z}$ is intractable for most practical models including deep neural networks: an analytic solution is lacking or may require an infeasible number of calculations.

Variational Inference (VI) techniques (Blei et al., 2017; Hoffman et al., 2013; Ranganath et al., 2014) provide a way to find an approximation of the posterior distribution. VI poses a family of distributions over latent variables $q(\mathbf{z}) \in \mathcal{Q}$ (e.g., Fig. 4b) and chooses the one that minimizes a chosen divergence[1] $D(q(\mathbf{z}) \parallel p(\mathbf{z}|\mathbf{x}))$ (e.g., Kullback-Leibler) to the true posterior distribution. VI often provides good approximations that can capture uncertainty, scaling to millions of data points using mini-batch training.

---

[1]Asymmetric distance

```python
def model(X, y=None):
  Wl = np.zeros((4, 3))
  Ws = np.ones((4, 3))

  bl = np.zeros(3)
  bs = np.ones(3)

  W = sample('W1', Normal(Wl, Ws))
  b = sample('b1', Normal(bl, bs))
  probs = softmax(X @ W + b)
  with plate('data', X.shape[0]):
    sample('y', Categorical(probs),
           obs=y)
```

```python
def guide(X, y=None):
  Wl = param('Wl', np.zeros((4, 3)))
  Ws = param('Ws', np.ones((4, 3)),
             constraint=positive)
  bl = param('bl', np.zeros(3))
  bs = param('bs', np.ones(3),
             constraint=positive)
  W = sample('W', Normal(Wl, Ws))
  b = sample('b', Normal(bl, bs))
```

(a) Probabilistic linear regression model $p(\mathbf{W}, \mathbf{b}, \mathbf{y})$

(b) Inference Program (guide) $q(\mathbf{W}, \mathbf{b})$ with parameters $\mathbf{W}_\ell$, $\mathbf{W}_s$, $\mathbf{b}_\ell$ and $\mathbf{b}_s$

```python
data = load_iris()
X_train, X_test, y_train, y_test = \
    train_test_split(data.data, data.target)

stein = Stein(model, WrappedGuide(guide), Adam(0.1), ELBO(),
              RBFKernel(), num_particles=100)
state, loss = stein.train(rng_key, 15000, X_train, y_train,
                          callbacks=[Progbar()])
```

(c) Running EinStein VI to learn parameters using Stein particles $\{\mathbf{W}_{\ell,i}, \mathbf{W}_{s,i}, \mathbf{b}_{\ell,i}, \mathbf{b}_{s,i}\}_{i=1}^{100}$

```python
y_pred = stein.predict(state, X_test)['y']
print(f"Accuracy: {np.mean(y_pred == y_test)*100:.2f} %")
```

(d) Using learned parameters from EinStein VI to predict values of test data

Figure 1: Linear regression example model in NumPyro with EinStein VI for inference

Stein Variational Inference (Liu and Wang, 2016) is a recent non-parametric approach to VI which uses a set of particles $\{\mathbf{z}_i\}_{i=1}^{N}$ as the approximating distribution $q(\mathbf{z})$ to provide better flexibility in capturing correlations between latent variables. The technique preserves the scalability of traditional VI approaches while offering the flexibility and modelling scope of techniques such as Markov Chain Monte Carlo (MCMC). Stein VI has been shown to be good at capturing multi-modality (Liu and Wang, 2016; Wang and Liu, 2019a), and has useful theoretical interpretations as particles following a gradient flow (Liu, 2017) and as a moment matching optimization system (Liu and Wang, 2018).

Many advanced inference methods based on Stein VI have been recently developed, including Stein mixtures (Nalisnick, 2019), non-linear Stein (Wang and Liu, 2019b), factorized graphical models (Zhuo et al., 2018; Wang et al., 2018b), matrix-valued kernels (Wang et al., 2019) and support for higher-order gradient-based optimization (Detommaso et al., 2018). These techniques have been shown to significantly extend the power of Stein VI, allowing more flexible and effective approximations of the true posterior distribution. While algorithmic power is growing, there remains a distinct lack of integration of these techniques into a general probabilistic programming language (PPL) framework. Such an integration would solve one of the most prominent limitations of traditional VI, which lacks flexibility to capture rich correlations in the approximated posterior.

This paper presents the EinStein VI library that extends the NumPyro PPL (Bingham et al., 2019; Phan et al., 2019) with support for the recent developments in Stein Variational Inference in an integrated and compositional fashion (see Fig. 1c and Fig. 1d). The library takes advantage of the capabilities of NumPyro— including universal probabilistic programming (van de Meent et al., 2018), integration with deep learning using JAX (Frostig et al., 2018), and automatic optimization and marginalization of discrete latent variables (Obermeyer et al., 2019)—to provide capabilities that work synergetically with the Stein algorithms. Concretely, our contributions are:

- Einstein VI, a general library that extends NumPyro and allows Stein Variational Inference to work with custom guide programs based on **ELBO-within-Stein** optimization (Nalisnick, 2019). The library is compositional with NumPyro features, including supported deep learning and automatic marginalization, loss functions (ELBO, Rényi ELBO, custom losses) and optimization methods, allowing it to grow organically with NumPyro development.

- Integration of recent developments in Stein variational inference within the EinStein library. This includes support for non-linear optimization (Wang and Liu, 2019b), a wealth of kernels (Liu and Wang, 2016; 2018; Gorham and Mackey, 2017), matrix-valued kernels (Wang et al., 2019) supporting higher-order optimization, and factorization based on conditional independence between elements in the model (graphical kernels).

- We support the application of transforms on the parameter space, including triangular (Parno and Marzouk, 2018) and neural transforms (Hoffman et al., 2018) to improve the gradient geometry of the inference problem.

- A series of examples demonstrate the power of an integrated library such as EinStein VI and the synergy between different Stein VI techniques. The examples include a novel Stein Mixture version of Deep Markov Models (SM-DMM), Stein Mixture Latent Dirichlet Allocation (SM-LDA), and several examples using neural transforms and higher-order optimization.

The paper proceeds as follows. We first present a primer on the theory of Stein VI in Section 2 relating it to our integrated implementation in EinStein VI. We discuss the general details of the implementation of EinStein VI in NumPyro in Section 3. We present the various examples using EinStein VI in Section 4 and finally summarize our results and future work in Section 5.

## 1.1 RELATED WORK

There has been a proliferation of deep probabilistic programming languages based on tensor frameworks with automatic differentiation, supporting various inference techniques.

Pyro (Bingham et al., 2019) is a universal PPL based on PyTorch (Paszke et al., 2019). The main mode of inference in Pyro is black-box Stochastic Variational Inference (Ranganath et al., 2014) with guides, which are flexible programs that approximate the posterior distribution by repeating the (non-observed) sample statements as the probabilistic model and can contain deep neural networks allowing amortization of inference (Kingma and Welling, 2014; Gershman and Goodman, 2014). Pyro also supports various sampling algorithms like Hamiltonian Monte Carlo/NUTS (Neal, 2011; Hoffman and Gelman, 2014) and sample-adaptive MCMC (Zhu, 2019) which provide more accurate posterior approximations but lack the scalability of VI techniques. NumPyro is a version of Pyro that runs on the JAX framework (Frostig et al., 2018), which allows it to exploit the powerful program optimization and parallelizability of the JAX compiler. Our library, EinStein VI, extends the Stochastic VI mode of inference by adding Stein Variational Inference on top of NumPyro, allowing the optimizable parameters in the guide to be approximated by a set of particles instead of a point estimate. Other languages with similar feature set include PyMC3 (Salvatier et al., 2016), Edward (Tran et al., 2016; 2019) and HackPPL (Ai et al., 2019).

## 2 PRIMER ON STEIN VI

The core idea of Stein VI (Liu and Wang, 2016) is to perform inference by approximating the target posterior distribution $p(\mathbf{z}|\mathbf{x})$ by an approximate distribution $q_{\mathcal{Z}}(\mathbf{z}) = N^{-1}\sum_i \delta_{\mathbf{z}_i}(\mathbf{z})$ based on a set of particles $\mathcal{Z} = \{\mathbf{z}_i\}_{i=1}^N$. Here, $\delta_{\mathbf{x}}(\mathbf{y})$ represents the Dirac delta measure, which is equal to 1 if $\mathbf{x} = \mathbf{y}$ and 0 otherwise. One could thus see the approximating distribution $q_{\mathcal{Z}}(\mathbf{z})$ as a mixture of point estimates, each represented by a particle $\mathbf{z} \in \mathcal{Z}$. The idea is then that we minimize the Kullback-Leibler divergence $D_{\mathrm{KL}}(q(\mathbf{z}) \parallel p(\mathbf{z}|\mathbf{x}))$ between the approximation and the true posterior by iteratively updating the particles using the Stein forces:

$$\mathbf{z}_i \leftarrow \mathbf{z}_i + \epsilon S_{\mathcal{Z}}(\mathbf{z}_i)$$

where $\epsilon$ represents the learning rate and $S_{\mathcal{Z}}$ the Stein forces.

**The Two Forces of Stein VI**   Stein VI consists of two forces which work additively under the form $S_{\mathcal{Z}} = S_{\mathcal{Z}}^+ + S_{\mathcal{Z}}^-$, where the attractive force

$$S_{\mathcal{Z}}^+(\mathbf{z}_i) = \mathbb{E}_{\mathbf{z}_j \sim q_{\mathcal{Z}}(\mathbf{z})}[k(\mathbf{z}_j, \mathbf{z}_i)\nabla_{\mathbf{z}_i} \log p(\mathbf{z}_i|\mathbf{x})]$$

and the repulsive force

$$S_{\mathcal{Z}}^-(\mathbf{z}_i) = \mathbb{E}_{\mathbf{z}_j \sim q_{\mathcal{Z}}(\mathbf{z})}[\nabla_{\mathbf{z}_i} k(\mathbf{z}_j, \mathbf{z}_i)].$$

Here $k : \mathbb{R}^d \times \mathbb{R}^d \to \mathbb{R}$ is a kernel. The attractive force can be seen as pushing the particles towards the direction that maximizes the true posterior distribution, smoothed by some kernel. For an example of a kernel, consider the standard RBF kernel $k(\mathbf{z}_i, \mathbf{z}_j) = \exp\left(-\frac{1}{h} \| \mathbf{z}_i - \mathbf{z}_j \|_2^2\right)$ with bandwidth parameter $h$, usually chosen as $\frac{1}{\log n}\mathrm{med}(\mathbf{z})$. The normalization constant becomes additive in the log-posterior $\log p(\mathbf{z}_i|\mathbf{x}) = -\log Z + \log p(\mathbf{x}|\mathbf{z}) + \log p(\mathbf{z})$ and so does not need to be computed for the gradient.

The repulsive force can be seen as moving particles away from each other, ensuring that they do not all collapse to the same mode. For the RBF kernel, the repulsive force becomes $\mathbb{E}_{\mathbf{z}_j \sim q_{\mathcal{Z}}(\mathbf{z})}[k(\mathbf{z}_j, \mathbf{z}_i)\frac{2}{h} \sum_\ell (\mathbf{z}_{i\ell} - \mathbf{z}_{j\ell})]$ and so particles that are close together (thus having a high kernel value) will be pushed away from each other.

**Non-linear Stein**   In non-linear Stein (Wang and Liu, 2019a), the repulsive force can be scaled by a factor $\lambda$, so $S_{\mathcal{Z}} = S_{\mathcal{Z}}^+ + \lambda S_{\mathcal{Z}}^-$ which is often useful when dealing with multi-modal distributions. It is also useful in our framework since the repulsive force often vanishes compared to the likelihood for large datasets $\mathcal{X}$ and therefore scaling the repulsive force by a constant $\lambda = \lambda(|\mathcal{X}|)$ proportional to the size of the dataset $|\mathcal{X}|$ appears logical, e.g. 0.1 or 0.01.

**Matrix-valued kernels**   The choice of kernels can be extended to matrix-valued ones (Wang et al., 2019) $K : \mathbb{R}^d \times \mathbb{R}^d \to \mathbb{R}^{d \times d}$ in which case the Stein forces become

$$S_{\mathcal{Z}}^+(\mathbf{z}_i) = \mathbb{E}_{\mathbf{z}_j \sim q_{\mathcal{Z}}(\mathbf{z})}[K(\mathbf{z}_j, \mathbf{z}_i)\nabla_{\mathbf{z}_i} \log p(\mathbf{z}_i|\mathbf{x})]$$

and

$$S_{\mathcal{Z}}^-(\mathbf{z}_i) = \mathbb{E}_{\mathbf{z}_j \sim q_{\mathcal{Z}}(\mathbf{z})}[K(\mathbf{z}_j, \mathbf{z}_i)\nabla_{\mathbf{z}_i}]$$

where the standalone del $\nabla_{\mathbf{z}_i}$ in the repulsive force represents the vector $\left(\frac{\partial}{\partial z_{i,1}}, \dots, \frac{\partial}{\partial z_{i,d}}\right)$ and so $(K(\mathbf{z}_j, \mathbf{z}_i)\nabla_{\mathbf{z}_i})_\ell = \sum_k \nabla_k K_{\ell,k}(\mathbf{z}_j, \mathbf{z}_i)$. The advantage of matrix-valued kernels is that they allow preconditioning using the Hessian or Fisher Information matrix (second-order derivatives), which can capture local curvature and thus achieve better optima than standard Stein VI. Furthermore, it is easy to capture graphical kernels (Wang et al., 2018b) $K = \mathrm{diag}(\{K^{(\ell)}\}_\ell)$ where the set of variables are partitioned with each their own local kernel $K^{(\ell)}$.

**ELBO-within-Stein**   The standard form of Stein VI guides the particles toward maximizing the log-posterior $\log p(\mathbf{z}|\mathbf{x})$ but in principle we could replace this with another objective (negative loss) $\mathcal{L}_{\boldsymbol{\theta}}$ which we want to maximize. Assume our guide $q_{\boldsymbol{\theta}}(\mathbf{z})$ comes from a parametric distribution with parameters $\boldsymbol{\theta}$, as in traditional VI. We can then for example maximize the ELBO (Kingma and Welling, 2014) $\mathcal{L}_{\boldsymbol{\theta}} = \mathbb{E}_{\mathbf{z} \sim q_{\boldsymbol{\theta}}(\mathbf{z})}[\log p(\mathbf{z}, \mathbf{x}) - \log q_{\boldsymbol{\theta}}(\mathbf{z})]$ as in traditional VI but using a set of particles for the parameters $\{\boldsymbol{\theta}_i\}_i$ guided by the Stein forces. One can see this as a generalization of traditional VI using a Stein-based mixture-model (Nalisnick, 2019), giving the parameters the flexibility to better capture correlations between distributions and to cover multiple modes (Wang and Liu, 2019a). Another perspective is that ELBO-within-Stein adds uncertainty to the approximating distribution $q_{\boldsymbol{\theta}}(\mathbf{z})$, allowing it to work with distributions over latent variables that are richer than simply point masses.

The objective to be maximized in our framework is compositional, which means that ELBO can be replaced by other objectives, e.g., Rényi ELBO (Li and Turner, 2016), Tail-adaptive f-divergence (Wang et al., 2018a) or Wasserstein pseudo-divergence (Ambrogioni et al., 2018) when these are implemented in NumPyro. Einstein VI makes it possible to integrate all these developments within Stein VI in a coherent fashion.

**Parameter Transforms**   Deterministic transport maps have been suggested to be integrated in MCMC methods by Parno and Marzouk (2018) and Hoffman et al. (2018), to allow constructing a better geometry for the posterior approximation and thus improve inference. We provide a novel adaptation of the idea to work with Stein-based methods, in the EinStein VI framework.

Consider the general objective we optimize in EinStein VI, which has the form:

$$S_\Theta(\boldsymbol{\theta}_i) = \mathbb{E}_{\boldsymbol{\theta}_j \sim q(\boldsymbol{\theta})}[k(\boldsymbol{\theta}_j, \boldsymbol{\theta}_i)\nabla_{\boldsymbol{\theta}_i}\mathcal{L}_{\boldsymbol{\theta}_i} + \nabla_{\boldsymbol{\theta}_i}k(\boldsymbol{\theta}_j, \boldsymbol{\theta}_i)]$$

where $\mathcal{L}_{\boldsymbol{\theta}}$ is our negative loss objective (e.g., ELBO) parameterised by $\boldsymbol{\theta}$ which we update using a set of Stein particles $\Theta = \{\boldsymbol{\theta}_i\}_i$.

The core idea is then that we would like to reparametrize the calculation so each Stein particle $\boldsymbol{\theta}_i$ is expressed as an (invertible) deterministic transformation of another particle $\boldsymbol{\phi}_i$, so $\boldsymbol{\theta}_i = \mathcal{T}(\boldsymbol{\phi}_i)$ (thus $\Theta = \{T(\boldsymbol{\phi})|\boldsymbol{\phi} \in \Phi\}$).

We do this reparametrization by first declaring the set of particles to be optimized $\Phi = \{\boldsymbol{\phi}_i\}_i$ and the transformation to be used $\mathcal{T}$, and then define the reparametrized Stein force as follows:

$$S_\Phi(\boldsymbol{\phi}_i) = S_\Theta(\mathcal{T}(\boldsymbol{\phi}_i))(\nabla_{\boldsymbol{\phi}_i}\mathcal{T}(\boldsymbol{\phi}_i))^\top$$

where $\nabla_{\boldsymbol{\phi}_i}\mathcal{T}(\boldsymbol{\phi}_i) : \mathbb{R}^{d \times e}$ is the reparametrization Jacobian, that adjusts for the change of variables (from $\boldsymbol{\theta}_i : \mathbb{R}^e$ to $\boldsymbol{\phi}_i : \mathbb{R}^d$).

The power of parameter transforms is amplified in our EinStein VI framework, in that we allow for its parameters to be learnable. We can thus use e.g., sparse triangular maps, invertible neural networks or normalizing flows, which have been shown to be particularly effective for Bayesian inference.

## 3   COMPOSITIONAL IMPLEMENTATION USING NUMPYRO

The system integrates well with the existing NumPyro API and programs using the standard Stochastic Variational Inference (SVI) interface can easily be converted to use EinStein VI instead. Fig. 1c shows the core usage of the EinStein VI interface; like the standard SVI interface it accepts a model, guide, optimizer and loss function. Additionally, EinStein VI accepts a kernel interface (e.g. `RBFKernel`) which is used to guide the Stein particles and to specify the number of Stein particles to use for inference.

We will discuss the major features of the implementation of Einstein VI, including allowing for re-initializable guides, the core algorithm that integrates Stein VI into NumPyro, and the compositional kernel interface.

### 3.1   RE-INITIALIZABLE GUIDES

The Stein VI interface requires that inference programs can be re-initialized with different values for each parameter. The reason is that we would like different Stein particles to be initialized differently in order for optimization to work correctly and avoid all particles collapsing into the posterior mode.

To support re-initializable guides we provide the `ReinitGuide` interface which requires implementing a function `find_params` that accepts a list of random number generator (RNG) keys in addition to the arguments for the guide and returns a set of freshly initialized parameters for each RNG key.

Most guides in NumPyro are written as functions as in Fig. 4b and to support those, we provide the `WrappedGuide` class which makes a callable guide re-initializable. works by running the provided guide multiple times, and reinitializing the parameters using NumPyro's interface as follows:

- `WrappedGuide` runs the guide transforming each parameter to unconstrained space.
- It replaces the values of the parameters with values provided by a NumPyro initialization strategy, e.g., `init_to_uniform`($r$) which initializes each parameter with a uniform random value in the range $[-r; r]$.
- It saves the parameter values for each particle and the required inverse transformations to constrained space, in order to allow running the model correctly.

We also allow parameters without reinitialization, e.g., for `stax` neural networks[2] which have their own initializers.

## 3.2 STEIN VI IN NUMPYRO

The integration of Stein VI into the deep PPL NumPyro requires handling transformations between the parameter representation of NumPyro—which is a dictionary mapping parameters to their values, which can be arbitrary Python types—to vectorized Stein particles that can be used in EinStein VI. For this, we rely on NumPyro's integration with JAX PyTrees[3] which can convert back and forth between Python collections and a flattened vectorized representation. For example, the parameters $\{\mathbf{W}_{\ell,i}, \mathbf{W}_{s,i}, \mathbf{b}_{\ell,i}, \mathbf{b}_{s,i}\}_i$ from Fig. 1 will be encoded into particles $\{\mathbf{a}_i\}_i$ where each particle is a 30-dimensional vector $\mathbf{a}_i \in \mathbb{R}^{30}$.

Alg. 1 shows the core algorithm of Einstein VI. Classical variational parameters $\phi$ and $\phi'$ are updated by averaging the loss over Stein particles. For Stein parameters, the process is more elaborate. First, we convert the set of individual parameters to set of vector-encoded particles using JAX PyTrees. Then we compute a kernel based on the vector-encoded Stein particles; the computation is kernel-dependent but an example is the RBF kernel which requires a bandwidth parameter $h$ that is dependent on the particle values (Liu and Wang, 2016). We apply JAX's `vmap` operator (Frostig et al., 2018; Phan et al., 2019) to parallely compute the Stein forces for each particle in a vectorized manner. Alg. 1 presents how EinStein VI works with regular-valued kernels but the actual implementation is more general, since it takes into account the different features presented in Section 2: non-linear repulsion force, potential use of matrix-valued kernels, and parameter transforms. Finally, we convert the monolithic Stein particle back to their non-vectorized dictionary-based form and return the expected changes for both classical and Stein parameters.

---

**Algorithm 1** EinStein VI

---

**Input:** Classical parameters $\phi$ and $\phi'$, Stein parameters $\{\boldsymbol{\theta}_i\}_i$, model $p_\phi(\mathbf{z}, \mathbf{x})$, guide $q_{\boldsymbol{\theta},\phi'}(\mathbf{z})$, loss $\mathcal{L}$, kernel interface KI.
**Output:** Parameter changes based on classical VI ($\Delta\phi$, $\Delta\phi'$) and Stein VI forces ($\{\Delta\boldsymbol{\theta}_i\}_i$).
$\Delta\phi \leftarrow \mathbb{E}_{\boldsymbol{\theta}}[\nabla_\phi \mathcal{L}(p_\phi, q_{\boldsymbol{\theta},\phi'})]$
$\Delta\phi' \leftarrow \mathbb{E}_{\boldsymbol{\theta}}[\nabla_{\phi'} \mathcal{L}(p_\phi, q_{\boldsymbol{\theta},\phi'})]$
$\{\mathbf{a}_i\}_i \leftarrow \text{PYTREEFLATTEN}(\{\boldsymbol{\theta}_i\}_i)$
$k \leftarrow \text{KI}(\{\mathbf{a}_i\}_i)$
$\{\Delta\mathbf{a}_i\}_i \leftarrow \text{VMAP}(\{\mathbf{a}_i\}_i, \mathbf{a}_i \mapsto \sum_{\mathbf{a}_j} k(\mathbf{a}_j, \mathbf{a}_i)\nabla_{\mathbf{a}_i}\mathcal{L}(p_\phi, q_{\text{PYTREERESTORE}(\mathbf{a}),\phi'}) + \nabla_{\mathbf{a}_i} k(\mathbf{a}_j, \mathbf{a}_i))$
$\{\Delta\boldsymbol{\theta}_i\}_i \leftarrow \text{PYTREERESTORE}(\{\Delta\mathbf{a}_i\}_i)$
**return** $\Delta\phi, \Delta\phi', \{\Delta\boldsymbol{\theta}_i\}_i$

---

## 3.3 KERNEL INTERFACE

The Kernel interface `SteinKernel` is reasonably simple. Users must implement the `compute` function, which accepts as input the current set of particles, the mapping between model parameters and particles, and the loss function $\mathcal{L}$ and returns a differentiable kernel $k$. Currently, all kernels are disallowed from changing state, but one could imagine a stateful interface in the future which relies on the history of particles, e.g., for computing conjugate gradients or quasi-Newton optimization. The list of implemented kernels in EinStein VI are described further in App. A.

## 4 EXAMPLES

We demonstrate the flexibility of EinStein VI by its application to two standard pathological examples (the Double Moon and Neal's Funnel), Stein Mixture Latent Dirichlet Allocation (SM-LDA), and Stein Mixture Deep Markov Model (SM-DMM). All experimental code, and the EinStein package are available at `https://github.com/aleatory-science/numpyro`. We plan to integrate our work upstream to the main NumPyro repository.

---

[2]`https://jax.readthedocs.io/en/latest/jax.experimental.stax.html`
[3]`https://jax.readthedocs.io/en/latest/notebooks/JAX_pytrees.html`

**Neal's funnel** We first consider a toy model where MCMC methods often struggle to sample the true posterior. The model was first described in Neal (2003), and is commonly known as Neal's funnel. The joint posterior is

$$y \sim \mathcal{N}(0, 3) \qquad x \sim \mathcal{N}(0, e^{\frac{y}{2}}),$$

which becomes difficult to sample as *y* grows increasingly negative. Figure 2 shows that EinStein VI captures the funnel well, and the particles are concentrated around the funnel area as expected.

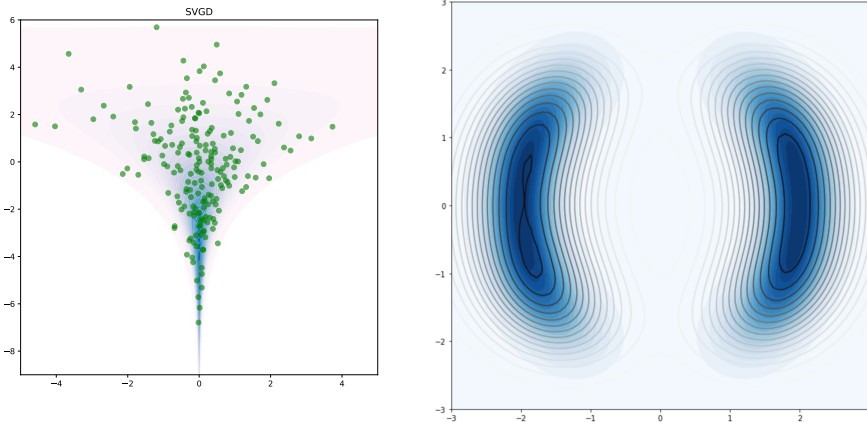

Figure 2: 100 particles fitted to Neal's Funnel using Stein in EinStein VI [left]. EinStein VI with neural transformation applied to the two moons model.

**Dual Moon** The Dual Moon example is a classical model [4] with multiple modes, which requires a rich family of variational distributions (guides) to provide a good approximation of the posterior. It thus serves as a good example of how integration of parameter transforms in EinStein VI.

In EinStein VI, each particle represents a variational parameter. Transformation of variables of the approximating distribution causes all particles to collapse at modes of the distribution. This is because the repulsive force is applied in the preimage of the transformation (e.g. Inverse Autoregressive flow, Planar flow, or Radial flow). To remedy this we apply Stein in the transformed space instead of using it to transform our approximation distribution. Figure 4 shows how to use neural flows for inference.

**Stein Mixture Latent Dirichlet Allocation** Latent Dirichlet Allocation (Blei et al., 2003) is a model that seeks to infer a distribution over topics $\boldsymbol{\theta}$, that represents a set of size $N$ of related words, from a collection of documents, each represented as a bag of words $\mathbf{w}$. Each document is generated according to the following model:

$$\boldsymbol{\theta} \sim \mathrm{Dir}(\alpha) \qquad \mathbf{z}_n \overset{\text{i.i.d}}{\sim} \mathrm{Cat}(\boldsymbol{\theta}) \qquad \mathbf{w}_n \overset{\text{i.i.d}}{\sim} \mathrm{Cat}(\varphi_{\mathbf{z}_n}) \qquad n \in \{1..N\}$$

where the number of topics $T = |\boldsymbol{\theta}|$ is fixed, $\alpha$ is a scalar that represents prior knowledge on the symmetric Dirichlet distribution, typically $\alpha = 1/T$ and $\{\varphi_t\}_{t=1}^{T}$ is a distribution of words for each topic. Especially interesting is that the model contains a set of discrete intermediate latent variables $\{\mathbf{z}_n\}_{n=1}^{N}$, which makes it non-differentiable per default. However, NumPyro supports automatic enumeration, which transforms the above generative model to the following equivalent model:

$$\boldsymbol{\theta} \sim \mathrm{Dir}(\alpha) \qquad \mathbf{w}_n \overset{\text{i.i.d}}{\sim} \sum_{t=1}^{T} \mathrm{Cat}(\varphi_t)\mathrm{Cat}(t|\boldsymbol{\theta}) \qquad n \in \{1..N\}$$

---

[4] https://github.com/pyro-ppl/numpyro/blob/master/examples/neutra.py

Figure 3

```
def flows(n_flows, in_dim, h_dim):      def guide(n_flows=3, h_dim=[2,2]):
  fs = []                                 flows = flows(n_flows, 2, h_dim)
  for i in range(n_flows-1):              particle = param('p', array([0.,0.]),
    arn = AutoRegNN(in_dim, h_dim,                        ComposeTrans(flows))
                out_fun=Elu)              numpyro.sample('x', Delta(particle))
    fs.append(IAutoRegTrans(arn))
    fs.append(PermTrans(
      arange(in_dim)[::-1]))
  arn = AutoregNN(in_dim, h_dim,
              out_fun=Elu)
  fs.append(IAutoRegTrans(arn))
  return fs
```

(a) Inverse AutoRegressive Transformation Flow.          (b) Guide with neural transformation

Figure 4: Neural transformed guide using Inverse Autoregressive Flows in Numpyro for EinStein VI (function names are shortened for formatting purposes).

Here, all the potential topic choices $t \in \{1..T\}$ are considered explicitly and marginalized (summed) out in the intermediate step. This way, we no longer have a discrete sampling step and we can differentiate the model again.

EinStein VI integrates with this automatic enumeration support in NumPyro, and thus provides more modelling power than traditional Stein methods that are not integrated into a corresponding powerful framework. In our example, the guide uses amortized inference (Gershman and Goodman, 2014) with a multi-layer perceptron (MLP) with 100 nodes and 20 potential topics, mapping each document to the distribution of topics, thus approximating the desired posterior $p(\boldsymbol{\theta}|\mathbf{w})$. For each Stein particle, we get a separate MLP, and thus the resulting distribution would be a Stein mixture of neural networks.

**Stein Mixture Deep Markov Model**   Music generation requires a model to learn complex temporal dependencies to achieve local consistency between notes. The Stein Mixture Deep Markov Model (SM-DMM) is a Deep Markov Model that uses a mixture of Stein particles to estimate distributions over model parameters. We consider a vectorized version of DMM (Jankowiak and Karaletsos, 2019) for the generation of polyphonic music using the JSB chorales dataset.

The SM-DMM model consists of two feed-forward neural networks. The *Transition* network transitions between latent states in the Markov chain and the *Emitter* produces an observation at each time step from the current latent state. The variational distribution is of the form $\prod_{n=1}^{N} q(z_{1:T_n}^n | f(X_{1:T_n}))$ where the parametrized feature function $f_{1:T_n}$ is a GRU (Chung et al., 2014), with hidden dimension 400. In the JSB chorales dataset, each time step is an eighth of a note.

We run the SM-DMM using the Adam optimizer (Kingma and Ba, 2014) with a learning rate of $10^{-5}$, using an RBF kernel and five Stein particles for one thousand epochs. The trained SM-DMM achieves a negative log likelihood of *6.67* on the test set, comparable to the 6.85 originally reported in Krishnan et al. (2016) and 6.82 reported in the improved version in Jankowiak and Karaletsos (2019).

## 5   SUMMARY

We presented EinStein VI, a novel library for Stein Variational Inference, built on top of NumPyro. It supports all recent techniques associated with Stein VI, including ELBO-within-Stein optimization, non-linear scaling of repulsive force and matrix-valued kernels, as well as adding novel features like parameter transforms. We have provided several examples that illustrate how EinStein VI has an easy-to-use high-level interface and performs well on pathological examples as well as Stein Mixture versions of realistic models such as Latent Dirichlet Allocation and Deep Markov Model.

In the future, we plan on supporting updates inspired by Stein Points (Chen et al., 2018; 2019), to handle cases where initialization of particles is sub-optimal, e.g. random restart of particles and MCMC updates to get better mode hopping properties.

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

ACKNOWLEDGEMENTS

We thank the Pyro development team for answering our questions regarding the Pyro implementation. We especially thank Martin Jankowiak for providing us guidance with and access to the Deep Markov Model example.

## A    KERNELS IN EINSTEIN VI

| Kernel | Definition | Detail | Type | Reference |
|--------|-----------|--------|------|-----------|
| Radial Basis Function (RBF) | $\exp(\frac{1}{h} \parallel \mathbf{x} - \mathbf{y} \parallel_2^2)$ | | scalar | Liu and Wang (2016) |
| | $\exp(\frac{1}{h}(\mathbf{x} - \mathbf{y}))$ | | vector | Pyro[5] |
| Inverse Multi-Quadratic (IMQ) | $(c^2 + \parallel \mathbf{x} - \mathbf{y} \parallel_2^2)^\beta$ | $\beta \in (-1, 0)$ and $c > 0$ | scalar | Gorham and Mackey (2017) |
| Random Feature Expansion | $\mathbb{E}_{\mathbf{w}}[\phi(\mathbf{x}, \mathbf{w})\phi(\mathbf{y}, \mathbf{w})]$ | $\phi(\mathbf{x}, \mathbf{w}) = \sqrt{2}\cos(\frac{1}{h}\mathbf{w}_1^\top \mathbf{x} + w_0)$ where $w_0 \sim \mathrm{Unif}(0, 2\pi)$ and $\mathbf{w}_1 \sim \mathcal{N}(0, 1)$ | scalar | Liu and Wang (2018) |
| Linear | $\mathbf{x}^\top \mathbf{y} + 1$ | | scalar | Liu and Wang (2018) |
| Mixture | $\sum_i w_i k_i(\mathbf{x}, \mathbf{y})$ | $\{k_i\}_i$ individual kernels and $w_i$ are weights | scalar, vector, matrix | Liu and Wang (2018) |
| Scalar-based Matrix | $k(\mathbf{x}, \mathbf{y})\boldsymbol{I}$ | $k$ scalar-valued kernel | matrix | Wang et al. (2019) |
| Vector-based Matrix | $\mathrm{diag}(k(\mathbf{x}, \mathbf{y}))$ | $k$ vector-valued kernel | matrix | Wang et al. (2019) |
| Graphical | $\mathrm{diag}(\{K^{(\ell)}(\mathbf{x}, \mathbf{y})\}_\ell)$ | $\{K^{(\ell)}\}_\ell$ matrix-valued kernels each for a unique partition of latent variables | matrix | Wang et al. (2019) |
| Constant Pre-conditioned | $\boldsymbol{Q}^{-\frac{1}{2}} K(\boldsymbol{Q}^{\frac{1}{2}}\mathbf{x}, \boldsymbol{Q}^{\frac{1}{2}}\mathbf{y})\boldsymbol{Q}^{-\frac{1}{2}}$ | $K$ is inner matrix-valued kernel and $\boldsymbol{Q}$ is a preconditioning matrix like the Hessian $-\nabla_{\bar{\mathbf{z}}}^2 \log p(\bar{\mathbf{z}}|\mathbf{x})$ or Fischer information $-\mathbb{E}_{\mathbf{z} \sim q_{\mathcal{Z}}(\mathbf{z})}[\nabla_{\mathbf{z}}^2 \log p(\mathbf{z}|\mathbf{x})]$ matrices | matrix | Wang et al. (2019) |
| Anchor Point Preconditioned | $\sum_{\ell=1}^m K_{\boldsymbol{Q}_\ell}(\mathbf{x}, \mathbf{y}) w_\ell(\mathbf{x}) w_\ell(\mathbf{y})$ | $\{\mathbf{a}_\ell\}_{\ell=1}^m$ is a set of anchor points, $\boldsymbol{Q}_\ell = \boldsymbol{Q}(\mathbf{a}_\ell)$ is a preconditioning matrix for each anchor point, $K_{\boldsymbol{Q}_\ell}$ is an inner kernel conditioned using $\boldsymbol{Q}_\ell$, and $w_\ell(\mathbf{x}) = \mathrm{softmax}_\ell(\{\mathcal{N}(\mathbf{x}|\mathbf{a}_{\ell'}, \boldsymbol{Q}_{\ell'}^{-1})\}_{\ell'})$ | matrix | Wang et al. (2019) |

