# OpenReview forum: "Einstein VI:   General and Integrated Stein Variational Inference in NumPyro"
_ICLR.cc/2021/Conference — Reject_

### Official Review · AnonReviewer1 · 2020-10-23
**Needs comparison to existing work**

**Rating:** 3
**Confidence:** 4

**Review:**

Unfortunately the authors link directly to the code, and the code is not anonymous. This might be a desk-reject as this is not a double blind review.

This work is a description of a library for developing variational inference algorithms using the ELBO-within-Stein framework developed in Nalisnick et al. (2017). The library is evaluated on on Neal's funnel and two moons, and on a polyphonic music dataset.

Comments

- Nalisnick et al was published in 2017. I assume this was a typo on the authors' part.

- Table A in the Appendix, describing different kernels, should include a column with computational and memory requirements for each kernel if they differ. This can affect the scalability.

- The work describes LDA but does not evaluate it. It would be helpful to include held-out log likelihood numbers on a standard topic modeling dataset such as 20 newsgroups. This would help people compare to prior work.

- Similarly, the library is evaluated by fitting to a standard polyphonic music dataset. Please report these numbers in a table, alongside a reasonable approach using standard variational inference and Stein VI (using the library) side-by-side. For example, the numbers here are much better, and use standard variational inference with the KL divergence: https://papers.nips.cc/paper/6039-sequential-neural-models-with-stochastic-layers.pdf (Stein Variational Inference can be difficult to understand, as can be Pyro, which is built on jax/pytorch, and the library developed here is built on top of all of these moving parts. Before embarking on using the library, a machine learning researcher should be very convinced that all this additional effort is worth it. Benchmarking this new library against existing work is important and will go a long way toward justifying its existence.)

- The references are very poorly formatted. Please clean up.

---

> ### Author Response · Authors · 2020-11-11
> **Code link**
>
> Dear Reviewer,
>
> Thanks for bringing it to our attention. I simply forgot to remove the link before submission, an honest mistake.
> I will make sure to be more thorough on next submission.

---

> ### Author Response · Authors · 2020-11-16
> **Response for Reviewer4**
>
> Thanks for the comments and feedback! We will certainly look into improving the lacking parts.

---

### Official Review · AnonReviewer4 · 2020-10-28
**A novel library for Stein VI, but the paper lacks details of the examples and evaluation**

**Rating:** 4
**Confidence:** 4

**Review:**

### Summary
This paper introduces EinStein VI: a lightweight composable library for Stein Variational Inference (Stein VI). The library is built on top of NumPyro and can take advantage of many of NumPyro's capabilities. It supports recent techniques associated with Stein VI, as well as novel features. The paper provides examples of using the EinStein VI library on different probabilistic models.

### Strengths
I'm not aware of other PPLs that support Stein Variational Inference. EinStein VI can provide an easier way to compare different Stein VI algorithms, and make research in the area easily reproducible.

### Concerns
The paper states that it provides examples that demonstrate that EinStein VI's interface is easy-to-use and performs well on pathological examples and realistic models. While it is true that there are several examples described, in my opinion there are not enough details to support the claims that EinStein VI is easy to use and performs well. A concrete comparison between EinStein VI and other methods is missing. It would have been helpful to have, for example, some concrete numbers (e.g. time taken to do inference, posterior predictive checks, posterior mean convergence plots, etc) that showcase why it is useful to use Stein VI for those examples, as opposed to other, already existing methods.

Another concern is that it is difficult to judge from the paper what the difference to (standard) NumPyro is. There is only a high-level explanation of the examples in the paper, so it's hard to imagine what the actual code looks like. Most importantly, I would have liked to see a comparison between EinStein VI code and what the code would have looked like without EinStein VI.

### Reasons for score
Unfortunately, there is not enough to go on in this paper, which is why I recommend reject. There is no strong evidence to support either the usability of the system (through elaborate examples and contrasting EinStein VI to other systems) or its performance (through experiments). This paper will be much stronger, and will have a better chance of reaching more people, if it includes either 1) more elaborate code examples that demonstrate that using EinStein is indeed better and easier than vanilla NumPyro, or 2) experiments comparing different Stein VI techniques to other inference algorithms, as evidence that a dedicated Stein VI library is indeed empowering our inference toolkit.

However, I do appreciate that writing a paper about tools / libraries is difficult, as the contribution of tools is typically a longer-term improvement in the workflow of developing new methods and techniques. I am open to increasing my score during rebuttal, depending on the answers of the questions listed below.

### Questions for the authors
Why has Stein VI not been implemented in PPL systems previously? Is it a matter of timing, or is there something particularly challenging about integrating Stein VI into a PPL?

The paper mentions "compositionality" several times. I was a little confused about what you mean by that: can you explain, perhaps with an example?

The paper mentions novel features (second to last paragraph page 8): can you elaborate?

The paper shows an example of using NeuTra in combination with Stein VI. Can you elaborate on the kind of problems that NeuTra won't be able to handle on its own? What about more lightweight approaches that can be applied in the context of probabilistic programming, such as "Automatic Reparameterisation of Probabilistic Programs" (Gorinova, Maria I., Dave Moore, and Matthew D. Hoffman. ICML 2020)? When will we see benefits of *both* applying a reparameterization that improves the posterior geometry, *and* using a more sophisticated inference algorithm like Stein VI?

### Suggestions for improvement and typos that have not affected the score I gave to the paper
Perhaps the most important change that would improve the paper is adding more concrete examples that would showcase the importance of using EinStein VI as opposed to simply NumPyro / other libraries. It would be nice to see a model where Stein VI gives us better inference results than a range of other algorithms / techniques and compare the code to what the user would have to write otherwise to achieve the same results. The examples of composing Stein VI with reparameterization / marginalization in NumPyro can be improved by comparing the results to Stein VI without reparameterization / marginalization and to other inference algorithms with reparameterization / marginalization.

Typos:
* last line of the abstract should be 500 000 as opposed to 500.000.
* URL in footnote 3 does not lead to the correct page

---

> ### Author Response · Authors · 2020-11-16
> **Response to Reviewer4**
>
> Thanks for the review and detailed comments! We appreciate the time and effort provided.
>
> We will add better evaluation examples and details for the next iteration. We will also be more concrete about performance and
>
> Answers to the question:
> * Basic SVGD support had already been implemented in Pyro, but the developers found the performance and flexibility wanting. We solved this issue
> * Compositionality means that it is possible to change the loss function (ELBO, RenyiELBO, etc.), kernel and inference program/guide without needing to rewrite the Stein interface. It is therefore easier to add new ideas if they do not significantly alter the main Stein algorithm!
> * The compositional implementation of the Stein features is in itself very novel, since all papers only mention features individually. The other novel feature is parameter transforms which AFAIK not been presented before.
> * Automatic Reparametrization is an orthogonal concern for out paper, but as far as I know it has been applied in the context of Pyro. It would be interesting to see whether one could port this to NumPyro (which I believe would be possible). In some sense, NumPyro is a very powerful framework in that all the advanced features can be combined, but it seems that there has been a lack of empirical investigation into its power :).

---

> > ### Comment · AnonReviewer4 · 2020-11-24
> > **Thank you for your response**
> >
> > Dear authors of paper 1105,
> >
> > Thank you for your response. I agree with AnonReviewer2 that the paper needs comparison to other methods. In addition, it is important that the paper highlights the importance of having a dedicated library for Stein inference through examples / a case study. Currently, there is not enough evidence in the paper to fairly judge the contribution at this moment.
> >
> > Regards,
> > AnonReviewer4

---

### Official Review · AnonReviewer3 · 2020-10-28
**Impressive implementation but very weak on theory and experimentation**

**Rating:** 5
**Confidence:** 4

**Review:**

Summary
========
The paper shows how a particle-based nonparameteric Variational Inference methodology known as Stein Variational Inference is integrated in a full-featured Probabilistic Programming Language, NumPyro. The paper goes into a fair amount detail describing a number of enhancements that have been made into numpyro using the general technique of particle-based representation of non-parameteric approximating distributions. They describe how geometric transforms of the parameter space can fit into their scheme, how matrix-valued kernels can be integrated. Also, they describe a new variant of Stein VI which they call ELBO-within-Stein. This introduces a new line of research for Stein VI. They also describe a Stein Mixture extension to Deep Markov Models (SM-DMM) and demonstrate on a very large dataset for the latter method.

Strengths
=========
- Integrating a more powerful variational approximation has clear benefits for probabilistic inference. And integrating this into a full-featured PPL allows users of Bayesian modeling to get access to a cutting edge technique with minimal programmatic effort.
- The integration of Stein VI into numpyro seems to have been very well designed given the very large number of ideas that have become easy to add including some innovative approaches.
- Showing state-of-the-art results on the high dimensional JSB-Chorales-dataset is a very impressive achievement for any PPL, and it certainly lends credence to this work.

Weaknesses
==========
- The only claim in the paper that is well supported is that the authors have extended NumPyro with SVI.
- The presentation style in the paper sometimes fails to draw a clear distinction between implementations of prior work in NumPyro versus new innovations. It is somewhat unclear whether the authors are making claims about the following points in their paper:
  * Non-linear Stein
  * Matrix-valued kernels
  * Parameter Transforms
  * Enumeration-based integration of discrete random variables
- The objective function of ELBO-within-Stein is not well motivated (see discussion below) and there is no direct comparison to the previous Stein Variational Gradient Descent which this method seeks to improve.
- There is no way to objectively evaluate the results on the first three experiments.

Recommendation
===============
Reject

Rationale
========
Experiments don't directly validate the main innovations of the paper.

Supporting Arguments
====================
- The main innovation in this paper appears to be the ELBO-within-Stein method. This appears to be different than SVGD (Stein Variational Gradient Descent). The difference appears to be that in the current paper both the entropy term and the Stein repulsion term are in the general objective (page 5 first equation) unlike in SVGD where the entropy term is not there. Philosophically, it doesn't look right to include both of these terms that are serving the same purpose (prevent the collapse of the variational approximation on the mode) . I could be mis-reading these equations, but if there are other difference the authors should clearly state and motivate these differences. Most importantly, the authors should show an experiment directly comparing to SVGD.
[SVGD reference: Qiang Liu and Dilin Wang.  Stein variational gradient descent: A general purpose Bayesianinference algorithm.Neural Information Processing Systems (NIPS), 2016.]
- The Neal's funnel should show the posterior marginal which is well known so that the reader can judge whether the samples are of good quality.
- Not clear how to interpret the dual moons plot. What are we looking at in this plot (right plot figure 2)? The posterior density or the true density? How do we know if this is a good posterior?
- For the LDA example there don't seem to be any results.

Questions for authors
===================
- Please provide motivation for the modification to SVGD objective.
- Please clearly state which of the many enhancements to NumPyro are being claimed as novel extensions.
- Any results worth sharing for the LDA?

Additional Suggestions Not Part of the Review Rating
=============================================
- The abstract mentions that this work is better than Stochastic VI but this claim is not actually supported explicitly. I had to read many of the referenced papers to realize that Jankowiak and Karaletsos (2019) had implemented a version of SVI. I'm assuming that this is what the abstract was referencing. Please do make such connections explicit!

- In Figure 1b, variables X and y are not actually used in the guide. Space permitting you could make a note as to why these are there in the guide.

- The first paragraph of the introduction mentions nuisance variables in Fig 4a. Not clear which variables in 4a were nuisance variables.

---

> ### Author Response · Authors · 2020-11-12
> **Response for Reviewer3**
>
> Thanks for the review of the paper and the comments!
>
> * Enumeration of discrete variables was already integrated in NumPyro before, while the other Stein-specific things we had to implement ourselves
> * We will try to better motivate the use of ELBO-within-Stein style objectives. For clarification, the objective is based Nalisnick 2017 work, but we make it more parametric w.r.t. choice of loss function. The current motivation is that one can add uncertainty to Stein particles moving from a mixture of point estimates to a mixture of distributions, and one can thus integrate it with Pyros support for custom inference programs (guides).
> * We will do more experiments on the LDA model and share the results.
>
> We will make the connection to SVI more explicit.

---

### Official Review · AnonReviewer2 · 2020-11-05
**Review for Einstein VI**

**Rating:** 5
**Confidence:** 3

**Review:**

In this paper, the authors developed a probabilistic programming framework for stein variational gradient descent and its variants using difference kinds of kernels, i.e. nonlinear kernels or matrix kernels. Simple experiments are included that the repository is effective and scalable for various problems.

Followings are a few of my questions and comments:

1. How is the new implementation compared with other frameworks using black box variational inference? For example, What is the speed of the training comparing with previous frameworks such as edward in large scale dataset tasks?  And the report does not give us a more thorough guide of the performance of each kernels for difference tasks.

2. The authors mentioned that the framework can be extended to use other objective function such as  Rényi ELB, , Tail-adaptive f-divergence, or Wasserstein pseudo-divergence. I am extremely confused about this part, since actually there is no objective function for svgd based methods (unless you design a new loss based on KSD or related things), how is this possible to combine other objective function using svgd? It would be great if the authors write down the derivations and have a detailed discussion.

3.  Does the current framework implement amortized svgd and other related stein's paper that can be utilized to train neural networks based applications such as stein-vae, stein-gan or kernel stein generative modeling [1, 2, 3]? This implementation can be important since it can be quite helpful for many other applications such as meta learning.

Also, the authors give the public code link of their implementation in the paper, which may expose their identity, but I am not sure if this violates anonymous requirement of ICLR submissions.

[1] Feng, Yihao, Dilin Wang, and Qiang Liu. "Learning to draw samples with amortized stein variational gradient descent." arXiv preprint arXiv:1707.06626 (2017).

[2] Wang, Dilin, and Qiang Liu. "Learning to draw samples: With application to amortized mle for generative adversarial learning." arXiv preprint arXiv:1611.01722 (2016).

[3] Chang, Wei-Cheng, et al. "Kernel Stein Generative Modeling." arXiv preprint arXiv:2007.03074 (2020).

---

> ### Author Response · Authors · 2020-11-12
> **Response for Reviewer2**
>
> Thanks for the review of the paper and the comments!
>
> * We have compared the performance of EinStein VI with standard Stochastic (Black-box) VI when running test examples, and the slow down is usually linear w.r.t. number of particles. We will consider adding more performance numbers in the next revision.
> * The idea is that we take any parametrized loss objective L(theta) and then infer a mixture of theta using Stein inference. We can elaborate more on this in the theory section in the next revision.
> * We can train VAEs, but there is a difference between our approach and amortized Stein. In our approach the whole neural network can be treated as a Stein particle, while as far as I understand, in amortized Stein one trains a neural network which one can sample particles from.
>
> We forgot about anonymizing the link to the code! Honest mistake.

---

> > ### Comment · AnonReviewer2 · 2020-11-23
> > **Thank the authors for the Response**
> >
> > I thank the authors for the response.
> >
> > Overall I think the authors should conduct more experiments and compare more related variational inference techniques on more large scale tasks to demonstrate the advantages. Overall, the current paper is not ready for publish and I tend to vote for rejection.

---

### Decision · Program_Chairs · 2021-01-07
**Final Decision**

**Decision:**

Reject

**Comment:**

All reviewers have carefully reviewed and discussed this paper. They are in consensus that this manuscript merits a strong revision. I encourage the authors to take these experts' thoughts into consideration in revising their manuscript.